# Risk of Pulmonary Fibrosis and Persistent Symptoms Post-COVID-19 in a Cohort of Outpatient Health Workers

**DOI:** 10.3390/v14091843

**Published:** 2022-08-23

**Authors:** Rosario Fernández-Plata, Anjarath-Lorena Higuera-Iglesias, Luz María Torres-Espíndola, Arnoldo Aquino-Gálvez, Rafael Velázquez Cruz, Ángel Camarena, Jaime Chávez Alderete, Javier Romo García, Noé Alvarado-Vásquez, David Martínez Briseño, Manuel Castillejos-López, Research Working Group

**Affiliations:** 1Department of Epidemiology and Statistics, National Institute of Respiratory Diseases “Ismael Cosío Villegas”, Tlalpan 4502, Mexico City 14080, Mexico; 2Laboratory of Pharmacology, National Institute of Pediatrics, Insurgentes Sur 3700, Mexico City 04530, Mexico; 3Laboratory of Molecular Biology of Emerging Diseases and COPD, National Institute of Respiratory Diseases “Ismael Cosío Villegas”, Tlalpan 4502, Mexico City 14080, Mexico; 4Genomics of Bone Metabolism Laboratory, National Institute of Genomic Medicine (INMEGEN), Mexico City 14610, Mexico; 5Laboratory of HLA, National Institute of Respiratory Diseases “Ismael Cosío Villegas”, Tlalpan 4502, Mexico City 14080, Mexico; 6Department of Bronchial Hyperreactivity, National Institute of Respiratory Diseases “Ismael Cosío Villegas”, Tlalpan 4502, Mexico City 14080, Mexico; 7Department of Biochemistry, National Institute of Respiratory Diseases “Ismael Cosío Villegas”, Tlalpan 4502, Mexico City 14080, Mexico

**Keywords:** SARS-CoV-2, pulmonary fibrosis, persistence of symptoms, post-COVID-19, outpatient

## Abstract

Background: Infection by SARS-CoV-2 has been associated with multiple symptoms; however, still, little is known about persistent symptoms and their probable association with the risk of developing pulmonary fibrosis in patients post-COVID-19. Methods: A longitudinal prospective study on health workers infected by SARS-CoV-2 was conducted. In this work, signs and symptoms were recorded of 149 health workers with a positive PCR test for SARS-CoV-2 at the beginning of the diagnosis, during the active infection, and during post-COVID-19 follow-up. The McNemar chi-square test was used to compare the proportions and percentages of symptoms between the baseline and each follow-up period. Results: The signs and symptoms after follow-up were cardiorespiratory, neurological, and inflammatory. Gastrointestinal symptoms were unusual at the disease onset, but unexpectedly, their frequency was higher in the post-infection stage. The multivariate analysis showed that pneumonia (HR 2.4, IC95%: 1.5–3.8, *p* < 0.001) and positive PCR tests still after four weeks (HR 5.3, IC95%: 2.3-12.3, *p* < 0.001) were factors associated with the diagnosis of post-COVID-19 pulmonary fibrosis in this study group. Conclusions: Our results showed that pneumonia and virus infection persistence were risk factors for developing pulmonary fibrosis post-COVID-19, after months of initial infection.

## 1. Introduction

COVID-19 is an infectious disease caused by the SARS-CoV-2 coronavirus, which was responsible for the outbreak in Wuhan, China, in December 2019 [1]. The identified symptoms caused by SARS-CoV-2 include cough, fever, dyspnea, headache, myalgia, arthralgia, odynophagia, chills, chest pain, rhinorrhea, anosmia, dysgeusia, and conjunctivitis [2,3]. Moreover, recently, balance disorders were reported in patients post-COVID-19; while olfactory and gustatory systems are affected particularly, vestibular and auditory symptoms were not significant [3]. Although most of these symptoms are usually mild in 80% of outpatients [4], some studies reported that 70–80% of patients who recovered from infection by the virus present one or more persistent symptoms [5] and that 1 to 5% of them progress to severe forms of the disease, requiring ventilatory support. Infections developing into severe conditions occur more frequently in older people or people with pre-existing chronic diseases [4,6], and recently, some reports suggest the probability of developing pulmonary fibrosis [5].

Most studies have focused on treating hospitalized or post-discharge patients [7,8,9,10,11,12]. Nonetheless, few studies have followed outpatients with a diagnosis previous to COVID-19, a reason why there is still insufficient knowledge about their sequels in this type of patient. Some studies have reported the persistence of at least one symptom after 125 days and neurological indicators after 98 days from the onset of symptoms in non-hospitalized patients [13,14]. Current evidence reported an impaired olfactory and gustatory system stemmed in patients post-COVID-19 after follow-up for six months [3]. Another study showed that 69% of non-hospitalized adults had to attend more than one outpatient visit after diagnosis during a follow-up period of 28–180 days, suggesting that health care is needed for a long time in these patients [15].

For that reason, the main objective of our work was to describe the clinical characteristics of ambulatory health workers infected previously with SARS-CoV-2 and to evaluate the possible association between persistent symptoms and the risk of developing pulmonary fibrosis after six months of initial infection.

## 2. Materials and Methods

### 2.1. Design and Study Population

A longitudinal prospective study was conducted from April 6 to December 14, 2021, at the National Institute of Respiratory Diseases “Ismael Cosío Villegas” (INER). All workers who presented clinical symptoms or had contact with a family member or co-worker who tested positive for the SARS-CoV-2 virus were considered for enrollment in this study. However, only workers with a positive viral-RNA test by RT-PCR were invited to participate. Those who agreed signed informed consent. We evaluated the clinical symptoms at three moments:At baseline, when workers attend to be tested for the first time.Three or five days after a positive RT-PCR test (active infection).Minimally, six months after a negative RT-PCR test (post-infection period).

The protocol and informed consent were approved by the INER Research, Bioethics, and Biosafety Committee. The approval number was E05–20.

### 2.2. Procedures

At baseline, all workers were tested and asked about their clinical symptoms. In

Mexico, all suspected cases of COVID-19 are registered in the Epidemiological Surveillance System for Respiratory Diseases (SISVER) of the Directorate General of Epidemiology (DGE). In the SISVER, risk factors, comorbidities, and initial symptoms detected as characteristic symptoms of the SARS-CoV-2 virus disease are recorded. A standardized questionnaire was designed to obtain the clinical symptoms from participants during all evaluation phases, but during the second and third measurements, the participants were reached by mobile phone. The questionnaire was applied through a Google drive document sent by an instant messaging application for smartphones. Patients were asked about the clinical information previously included in the Epidemiological Surveillance System for Respiratory Diseases (SISVER), and variables associated with new symptoms were added. Additionally, the symptoms were classified into four categories: neurological, gastric, inflammatory, and cardiorespiratory.

### 2.3. Post-COVID Assessment of Pulmonary Fibrosis

Although CT is not considered within the standard treatment, some evidence [5] suggested the probability of developing pulmonary fibrosis in these patients. This gives support to deciding to perform the CT three or five days after a positive RT-PCR test and six months after a negative RT-PCR test in these patients. According to 2015 consensus diagnostic guidelines for PF by the American Thoracic Society (ATS), pulmonary fibrosis was defined as a combination of tomographic findings, including parenchymal bands, irregular interfaces, a thick reticular pattern, and bronchiectasis, which was confirmed by the clinical assessment of a pulmonologist.

### 2.4. Statistical Analysis

Categorical variables were presented as frequency rates and percentages, and quantitative variables were described using the median and interquartile range. Characteristics of the study population were described for each measurement. We used McNemar’s chi-square to compare proportions among measurements. The univariate and multivariate Cox proportional hazards analyses were used to assess the independent prognosis factors for post-COVID-19 pulmonary fibrosis. A result was considered statistically significant if its CI95% did not include the null value. Statistical analysis was performed using STATA software (version 14; Stata Corp, College Station, TX, USA).

## 3. Results

### 3.1. Group of Study

According to international guidelines, 288 health workers were confirmed positive for SARS-CoV-2. One hundred twenty-five were excluded from participating in this study, leaving one hundred sixty-three eligible participants who gave their consent to enroll. Later, 14 more left the study after the first or second measurement, leaving a final study population of 149 patients (Figure 1). Of the 149 health workers who met the inclusion criteria, 63% were female. The study population’s age median (p25, p75) was 35 years (29, 45). Sixty-two percent of the participants were medical personnel: 14% administrative staff, and 24% came from different hospital areas. Forty-six (31%) of the patients reported three or more symptoms at the beginning of the study.

### 3.2. Clinical Characteristics

A total of 17% of the participants indicated at least one comorbidity, among which obesity was the most frequent with 5.4%, followed by hypertension with 4.7%, and diabetes with 3.4% (Table 1). In addition, 4.7% of patients were smokers, 95.3 % had the BCG vaccine, while 81.2% had received the influenza vaccine. A total of 58 patients (38.9%) presented pneumonia during the first 7–10 days of infection, of which 21 scored 2 points and the rest 1 point according to the CURB65 scale. Regarding the elapsed days from a positive to a negative PCR test, the median (p25, p75) was 17 (15, 30), while the median (p25, p75) from a positive test to the last questionnaire applied was 128 days (99, 153).

### 3.3. Signs and Symptoms Prevalence

Signs and symptoms reported by the patients showed high variability in the three times they were evaluated. The most frequent baseline symptoms were headache (43.6%), cough (38.9%), and rhinorrhea (26.9%), followed by arthralgia (22.2%) and fever (20.8%). While in active infection, fatigue or weakness were the most frequent symptoms (65.1%), followed by headache (59.7%), anosmia (55.0%), myalgia (53.7%), and dysgeusia/ageusia (51.0%), principally; however, a great number of other symptoms was described too (e.g., arthralgia, odynophagia, sleeping problems, etc.) (Table 2). During the post-infection period, fatigue (36.9%) and dyspnea (22.8%) were the most prevalent symptoms, followed by headache (21.5%) and hair loss (21.5%) (Table 2). Comparing baseline reports versus active infection, the highest increments over the prevalence of signs and symptoms were fatigue (0.7 vs. 65%, *p*-value < 0.0001), anosmia (4.7 vs. 55%, *p*-value < 0.0001), and dysgeusia/ageusia (5.4 vs. 51%, *p*-value < 0.0001).

Additionally, the unreported baseline symptoms that manifested during active infection and persisted during post-infection (Table 2) were:(1)Neurological: fatigue or weakness, difficulty concentrating, blurred vision, hair loss, cramps, ear disorders, sleeping problems, and anxiety;(2)Gastric: bite alteration and weight changes;(3)Inflammatory and cardiorespiratory: dermatitis, dyspnea, and tachycardia.

Statistical analysis showed significant differences in all cases (1 vs. 2 or 3; *p* < 0.0001) (Table 2).

### 3.4. Diagnosis of Pulmonary Fibrosis and Risk Factors for the Development of Post-COVID-19 Pulmonary Fibrosis

Pulmonary fibrosis was diagnosed in 31/149 (21%) patients based on a combination of tomographic findings, including parenchymal bands, irregular interfaces, a thick reticular pattern, and bronchiectasis confirmed by a pulmonologist. Figure 2 and Figure 3 shows computed tomography scans from patients without (Figure 2) or with (Figure 3) diagnosis of pulmonary fibrosis after a follow-up of six months from initial infection by SARS-CoV-2.

The univariate Cox regression analysis showed that patients with pneumonia had a significantly higher risk of developing post-COVID-19 pulmonary fibrosis than those without pneumonia (hazard ratio (HR) 2.2, IC95%: 1.4–3.5, *p* = 0.0007). Similarly, patients with positive PCR test > 4 weeks had a significantly higher risk than patients without pneumonia (HR 4.4, IC95%: 2.1–8.7, *p* < 0.001). In comparison, age (HR 1.01, IC95%: 0.97–1.05, *p* < 0.46) and sex (HR 1.2, IC95%: 0.59–1.99, *p* < 0.55) did not show statistically significant differences.

The univariate analysis included both comorbidities and vaccination status. However, we decided not to include them in the multivariate model to maintain parsimonious models that permit the prediction of fibrosis. The confounding factors such as sex and age were adjusted by Cox regression despite having *p*-values greater than 0.10 (Table 3).

## 4. Discussion

This study is the first prospective cohort with an extended follow-up period to assess symptom persistence and risk of developing post-COVID-19 pulmonary fibrosis in outpatient healthcare workers previously infected by the SARS-CoV-2. Our work contributes to the generation of knowledge since it shows the consequences of acute illness in ambulatory health personnel six or more months after the apparition of the first symptoms. Results showed that a significant proportion of patients presented persistent symptoms of the nature of cardiorespiratory, neurological, gastrointestinal, or inflammatory, which were reported since the early stages of the disease and maintained at the six-month follow-up. In addition, some signs and symptoms did not show statistically significant differences between the initial and final assessment, indicating that these symptoms persist after the initial infection.

Our data also show that respiratory symptoms prevailed in most patients. This finding is important because it is one of the few viral infections that cause a high percentage of long-term persistent respiratory symptoms in outpatients. Respiratory sequelae have been commonly observed in hospitalized patients with severe influenza but rarely in other infections [16]. Several studies have reported the persistence of neurological symptoms in patients with COVID-19 that can manifest as a neurological syndrome with diverse and complex characteristics [17]. Among these neurological manifestations, the most common are headaches, dizziness, loss of taste and smell, encephalitis, and cerebrovascular disease [2,3,18]. Central and peripheral nervous system disorders associated with acute COVID-19 are usually transient; however, neurological sequelae may persist for months [19]. In this sense, whether the chronic sequelae can become reversible remains without a definitive answer and is one question that can be answered with prolonged follow-up of these patients.

RNA from the SARS-CoV-2 virus has been identified in stool samples from infected patients [20], and the viral receptor is expressed on gastrointestinal epithelial cells, indicating that this virus can actively infect and replicate in the gastrointestinal tract [21], even without developing gastrointestinal symptoms [22]. Recent evidence shows that the incidence rate of vomiting and diarrhea found by us during infection is similar to that reported by other studies [23]. However, the ageusia/dysgeusia ratio was higher than that reported in a systematic review by Mehraeen [24]. Here, we show that gastrointestinal symptoms are rare at disease onset in outpatients with COVID-19, but surprisingly, their frequency was higher in the post-infection stage of the disease.

It is essential to note several persistent inflammatory symptoms in this study. As far as we know, the mechanisms that lead to the persistence of inflammatory symptoms post-COVID are unknown. However, like complications in other organs or systems, cell damage and a robust innate immune response associated with the production of inflammatory cytokines can contribute to the development of this persistence. The risk of persistent symptoms such as conjunctivitis, dermatitis, and hair loss in the post-acute phase of COVID-19 is likely related to the duration and severity of a hyper-inflammatory state, although how long it persists is unknown. More research is needed to provide insight into the immunological mechanisms of the disease.

In this context, it could be assumed that there is a very high probability that there will be a transition from acute cases of COVID-19 to cases of patients with post-COVID sequelae, which could associate with the development of pulmonary fibrosis. Therefore, it is appropriate to focus on the medium- and long-term consequences in recovered patients since it is becoming a public health problem, which will require current and future multidisciplinary efforts to treat these patients. Our study agrees with other works, where a proportion of critically ill patients who recovered from pneumonia associated with the COVID-19 infection were reported to have early lung damage, which derived from the chronic deterioration of lung function and its evolution to pulmonary fibrosis [25]. However, despite mentioning and suggesting the probable development of fibrosis in patients with COVID-19, the presence of pulmonary fibrosis in outpatients has still rarely been studied [5,25]. In our study, 21% of the cases had fibrosis as a severe sequela.

Although previous works had suggested the possibility of developing pulmonary fibrosis in post-COVID patients, our results were unexpected. The above underlines the importance of follow-up with these patients, whereby our next objective is to extend its monitoring period to at least 12 months. Pneumonia at the beginning of the acute condition and positivity in the PCR test for SARSCoV2 for four weeks or more were risk factors for the fibrosis-associated damage. Our findings are consistent with previous studies, which indicate that mild SARS-CoV-2 disease has multi-systemic repercussions [26] and show the importance of characterizing symptoms after the COVID-19 acute stage and compared to the symptoms in the first days of the infectious disease. At this point, fibrosis has been reported to be a long-term sequela in the lungs of patients who have recovered from COVID-19 infection [27]. Likewise, it is alarming that recent evidence shows that post-COVID fibrosis can occur regardless of age or comorbidities in patients [28]. In this regard, Li et al. [29] recently reported the development of pulmonary fibrosis in old patients with higher BMI, in critical condition, with longer viral clearance time, and with delayed hospitalization. Compared with our work, the similitude is in the persistence of viral infection; however, our patients were outpatients without a critical condition or extended hospitalization. The above highlights the importance of follow-up for patients with a diagnosis of COVID-19 independently of its clinical evolution. Likewise, the importance of inflammatory components, which were independent of the severity of the disease, have been reported, but were associated with an increase in the relative risk of developing lung fibrosis-like changes [30]. This last item is an element to consider as a risk factor in patients post-COVID-19, independently of the clinical condition of patients. However, although the inflammatory response and coagulopathies stated post-infection by the SARS-CoV-2 have been associated with the development of pulmonary fibrosis, only some patients develop fibrosis, which makes the landscape even more complex [31]. Additionally, based on the recent findings of other authors, another possibility is that COVID-19 sequelae could result from hemostatic alterations due to the vascular endothelium damage and the intense inflammatory response, which leads to multi-systemic damage, and probably more, after the onset of pulmonary fibrosis [32].

## 5. Limitations

This study has an epidemiological rather than a clinical background, so caution must be taken to interpret these findings. Due to the nature of the data collection (indirect survey), the patient may have reported some incorrectly described symptoms. However, the three measurements maximized the study’s statistical power and reduced the risk of making a type II error.

## 6. Conclusions

This study highlights the high frequency of respiratory symptoms after COVID-19 and the possible association of initial pneumonia and a persistently positive test after four weeks as risk factors for post-COVID-19 pulmonary fibrosis. Additionally, our results underline the importance of follow-up with patients over more time to reduce or attenuate the possibility of developing pulmonary fibrosis. However, although more studies are required to confirm them, the probable development of fibrosis in patients post-COVID-19 independent of age, co-morbidities, or symptoms highlights the importance of this issue to future works.

## Figures and Tables

**Figure 1 viruses-14-01843-f001:**
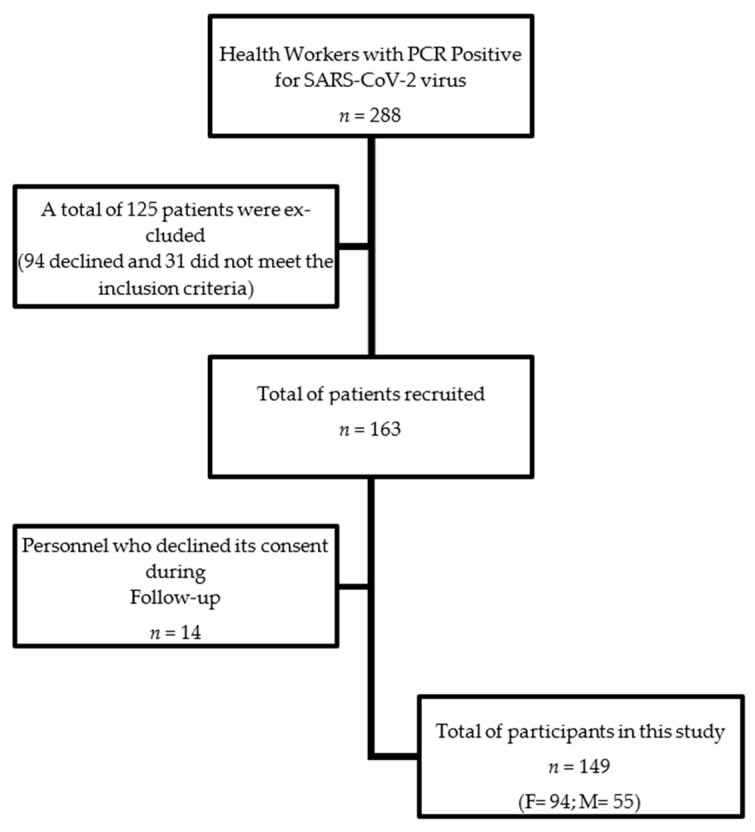
Flow chart of Study Population (F = Female; M = Male).

**Figure 2 viruses-14-01843-f002:**
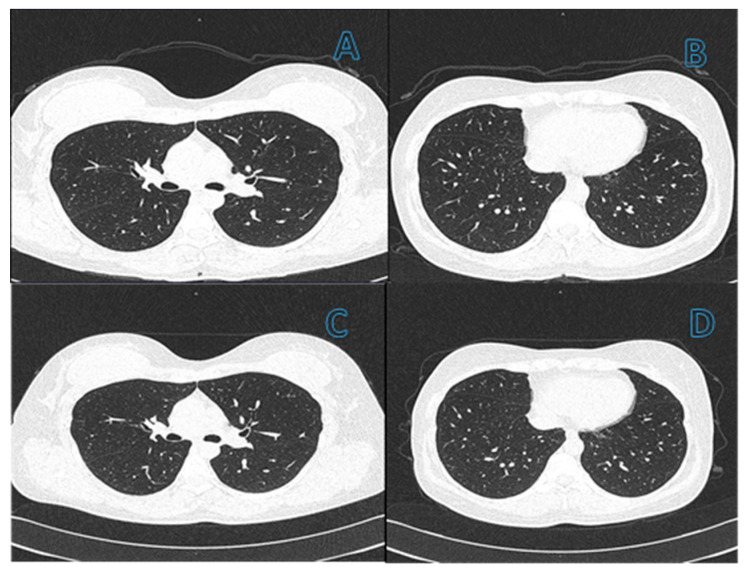
(**A**,**B**) Computed tomography (CT) scans of the chest with 1 mm slices of a 46-year-old female patient in the active phase of infection. CT does not show changes in the lung related to pneumonia. (**C**,**D**) Images of CT six months after resolution of SARS-CoV-2 infection have no change from initial CT ones, which discard a history of lung damage.

**Figure 3 viruses-14-01843-f003:**
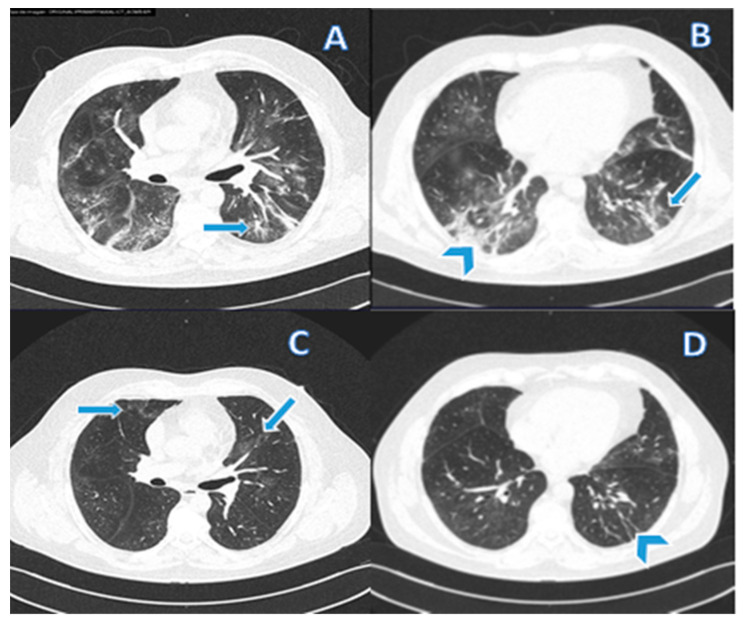
(**A**,**B**) CT from a 56-year-old male patient in the active infection phase of COVID-19. (**C**,**D**) CT after six months of infection resolution. Typical findings of COVID-19 pneumonia and tomographic findings that support the diagnosis of pulmonary fibrosis are observed. (**A**) Vascular thickening (arrow) associated with an area of ground-glass opacity; (**B**) Subpleural parenchymal bands (arrows) and ground-glass opacity and consolidation (arrowheads); (**C**) Absorption of most of the affected areas leaving some lesions in ground glass; and (**D**) Fibrous lesions that represent residual organizing pneumonia.

**Table 1 viruses-14-01843-t001:** Clinical characteristics and comorbidities were initially reported by the patients evaluated.

Variable	*n* (%)
Female	91 (63.0)
Age	35 (29–45) *
Obesity	8 (5.4)
Hypertension	7 (4.7)
Diabetes	5 (3.4)
Smoking	7 (4.7)
BCG vaccine	142 (95.3)
Influenza vaccine	121 (81.2)
Pneumonia	58 (38.9)

* Median and interquartile range.

**Table 2 viruses-14-01843-t002:** Signs and symptoms prevalence in the study population (*n* = 149) reported at baseline, in active infection, and during follow-up by six months.

	Baseline (1)	Active Infection (2)	Post-Infection (3)	1 vs. 2	1 vs. 3	2 vs. 3
Signs and Symptoms	(%)	(%)	(%)	*p*-Value	*p*-Value	*p*-Value
**Neurological**						
General attack	9.4	28.2	3.4	<0.0001	0.064	<0.0001
Arthralgia	22.2	42.3	14.8	<0.0001	0.093	<0.0001
Myalgia	14.1	53.7	13.4	<0.0001	0.999	<0.0001
Dysgeusia/Ageusia	5.4	51.0	10.7	<0.0001	0.115	<0.0001
Anosmia	4.7	55.0	13.4	<0.0001	0.011	<0.0001
Odynophagia	9.4	40.3	7.4	<0.0001	0.664	<0.0001
Abdominal pain	4	18.8	6.0	<0.0001	0.581	<0.0001
Headache	43.6	59.7	21.5	0.005	<0.0001	<0.0001
Fatigue or weakness	0.7	65.1	36.9	<0.0001	<0.0001	<0.0001
Difficult concentrating	0	29.5	14.8	<0.0001	<0.0001	0.015
Blurred vision	0	8.7	8.1	<0.0001	<0.0001	0.617
Hair loss	0	24.2	21.5	<0.0001	<0.0001	0.999
Cramps	0	15.4	12.1	<0.0001	<0.0001	0.774
Ear disorders	1.3	26.2	12.1	<0.0001	<0.0001	0.001
Sleeping problems	0	34.9	17.5	<0.0001	<0.0001	<0.0001
Anxiety	0	20.8	14.1	<0.0001	<0.0001	0.041
**Gastric**						
Diarrhea	6.7	33.6	6.0	<0.0001	0.999	<0.0001
Vomit	0	7.4	0	0.001	0.999	0.001
Nausea	1.3	16.1	2.0	<0.0001	0.999	<0.0001
Xerostomia	0.7	26.9	7.4	<0.0001	0.006	<0.0001
Mouth ulcers	0	6.7	3.4	0.002	0.063	0.227
Bite alteration	0	21.5	13.4	<0.0001	<0.0001	0.012
Weight changes	0	14.1	10.7	<0.0001	<0.0001	0.808
**Inflammatory**						
Conjunctivitis	5.4	20.8	4.7	<0.0001	0.999	<0.0001
Lymphadenopathy	0	17.5	3.4	<0.0001	0.063	<0.0001
Dermatitis	0	16.8	12.1	<0.0001	<0.0001	0.167
Irritability	4.0	22.8	7.4	<0.0001	0.332	<0.0001
Diaphoresis	10.7	42.3	10.1	<0.0001	0.999	<0.0001
Fever	20.8	36.2	0	0.001	<0.0001	<0.0001
**Cardiorespiratory**						
Rhinorrhea	26.9	27.5	4.7	0.882	<0.0001	<0.0001
Nasal congestion	1.3	27.5	6.7	<0.0001	0.039	<0.0001
Cough	38.9	40.3	7.4	0.773	<0.0001	<0.0001
Dyspnea (mild/moderate)	6.0	40.3	22.8	<0.0001	<0.0001	0.001
Tachycardia	0	28.2	14.8	<0.0001	<0.0001	0.001

**Table 3 viruses-14-01843-t003:** Univariate and multivariate hazard ratios for post-COVID-19 pulmonary fibrosis.

	Univariate			Multivariate		
Variable	HR	CI 95%	*p*-Value	HR	CI 95%	*p*-Value
Pneumonia	2.2	1.4–3.5	0.0007	2.41	1.51–3.82	<0.001
PCR positive test > 4 weeks	4.4	2.1–8.7	<0.0001	5.38	2.34–12.35	<0.001
Age	1.01	0.97–1.05	0.46	0.99	0.95–1.04	0.96
Sex	1.2	0.59–1.99	0.55	1.3	0.6–1.8	0.47
Diabetes	0.94	0.29–3.05	0.93	-	-	-
Hypertension	1.28	0.46–3.54	0.63	-	-	-
Obesity	1.9	0.69–5.28	0.23	-	-	-
Smoking	1.77	0.71–0.44	0.21	-	-	-
BCG vaccine	0.69	0.42–1.21	0.2	-	-	-
Influenza vaccine	0.91	0.91–2.33	0.68	-	-	-

Abbreviations: HR, hazard ratio; CI, confidence interval; PCR, polymerase chain reaction.

## Data Availability

Data can be obtained from the corresponding author upon reasonable request.

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
