# Peer review of "Risk of Pulmonary Fibrosis and Persistent Symptoms Post-COVID-19 in a Cohort of Outpatient Health Workers"

_viruses, 2022, doi:10.3390/v14091843_

Round 1

Reviewer 1 Report

It is a work of current interest, which adds value to the medical practice applied in the management of patients post COVID 19. Also, in my opinion, I believe that their follow-up period should be extended for months 9 and 12, considering the persistence of symptoms . An evaluation of the evolution of pulmonary fibrosis is also necessary, in addition to imaging, such as a molecular evaluation of the markers associated with fibrosis as well as an evaluation of the chronic inflammatory status.

Author Response

Thank you for your time and comments on our manuscript.

Comments and Suggestions for Authors

R: It is a work of current interest, which adds value to the medical practice applied in the management of patients post-COVID 19. Also, in my opinion, I believe that their follow-up period should be extended for months 9 and 12, considering the persistence of symptoms. An evaluation of the evolution of pulmonary fibrosis is also necessary, in addition to imaging, such as a molecular evaluation of the markers associated with fibrosis as well as an evaluation of the chronic inflammatory status.

A: In relation to your comment and based on the results obtained, our next objective is to extend the follow-up period of these patients to at least 12 months. Taking into consideration that we only started from a hypothesis that suggested the probability of developing pulmonary fibrosis, our results were unexpected. However, we are now considering evaluating other markers that allow us to do a better clinical follow-up of post-covid patients.

Additionally, now included in the Discussion and conclusions sections, the next paragraphs:

Discussion:

Although previous works had suggested the possibility of developing pulmonary fibrosis in post-COVID patients; our results were unexpected. The above underlines the importance of follow-up with these patients, whereby our next objective is to extend its monitoring period to at least 12 months.

Conclusion:

Also, our results underline the importance of follow-up of patients for more time, to reduce or attenuate the possibility of developing pulmonary fibrosis.

Reviewer 2 Report

Dear authors, Dear Editors

I read with interest the manuscript of Fernandez Plata et al. on post-covid persistant symptoms and pulmonary fibrosis.
authors prospectively assess symptoms and parenchymal alterations after covid-19 infection in a cohort of healthcare workers with low prevalence of comorbidities.

Several remarks:

- all subjects benefited from a pulmonary CT scan at 6 months: this was not part of international guidelines nor the usual clinical practice unless symptomatic; If the CT scan was part of the study protocol it should be mentioned as such.

- the paper was focussed on symptoms but other factors could be implicated in the pulmonary fibrosis genesis such as the presence of pulmonary pathology at baseline (COPD, asthma etc). Moreover tabacco consumption was not included in the univariate analysis, nor the cardiovascular risk factors.
Considering the univariate and multivariate analysis, I think it would be better to include both in your tables. Indeed table 3 is entitled 'univariate and multivariate analysis" but it only shows the multivariate analysis result. It is not clear on which criteria variables were included in the multivariate analysis (which p limit?). this should also appear in the statistical analysis chapter. Moreover, for the uni/multi variate analysis, OR rather than HR should appear.

Author Response

Thank you for your time and comments on our manuscript.

R: English language and style are fine/minor spell check required.

A: The English of the manuscript was revised according to your comment.

Several remarks:

R: all subjects benefited from a pulmonary CT scan at 6 months: this was not part of international guidelines nor the usual clinical practice unless symptomatic; If the CT scan was part of the study protocol it should be mentioned as such.

A: Although CT is not considered within the standard treatment for these patients, some evidence indicated that it was very likely that some patients could develop pulmonary fibrosis, a reason which is it was decided to perform the CT on these patients.

The next paragraph is now included in the section of Materials and Methods:

Although CT is not considered within the standard treatment, some evidence [5] suggested the probability of developing pulmonary fibrosis in these patients. This gives support to deciding to perform the CT three or five days after a positive RT-PCR test, and six months after a negative RT-PCR test in these patients.

R: The paper was focussed on symptoms but other factors could be implicated in the pulmonary fibrosis genesis such as the presence of pulmonary pathology at baseline (COPD, asthma etc). Moreover tabacco consumption was not included in the univariate analysis, nor the cardiovascular risk factors. Considering the univariate and multivariate analysis, I think it would be better to include both in your tables. Indeed table 3 is entitled 'univariate and multivariate analysis" but it only shows the multivariate analysis result. It is not clear on which criteria variables were included in the multivariate analysis (which p limit?). this should also appear in the statistical analysis chapter. Moreover, for the uni/multivariate analysis, OR rather than HR should appear.

A: According to your comments the next paragraph and a new Table 3 are now included: The univariate analysis included both comorbidities and vaccination status. However, we decided not to include them in the multivariate model to maintain parsimonious models which permit the prediction of fibrosis. The confounding factors such as sex and age were adjusted by Cox regression despite having p-values greater than 0.10 (Table 3).